# A Mathematical Programming Approach for IoT-Enabled, Energy-Efficient Heterogeneous Wireless Sensor Network Design and Implementation

**DOI:** 10.3390/s24051457

**Published:** 2024-02-23

**Authors:** Ertugrul Taparci, Kardelen Olcay, Melike Ozlem Akmandor, Banu Kabakulak, Baykal Sarioglu, Yigit Daghan Gokdel

**Affiliations:** Faculty of Engineering and Natural Sciences, Istanbul Bilgi University, 34060 Istanbul, Türkiye; kardelen.olcay@bilgiedu.net (K.O.); melike.akmandor@bilgiedu.net (M.O.A.); banu.kabakulak@bilgi.edu.tr (B.K.);

**Keywords:** internet of things, wireless sensor network, narrow-band communication, energy efficiency, mathematical programming, optimization, data loss avoidance, smart farming

## Abstract

The Internet of Things (IoT) is playing a pivotal role in transforming various industries, and Wireless Sensor Networks (WSNs) are emerging as the key drivers of this innovation. This research explores the utilization of a heterogeneous network model to optimize the deployment of sensors in agricultural settings. The primary objective is to strategically position sensor nodes for efficient energy consumption, prolonged network lifetime, and dependable data transmission. The proposed strategy incorporates an offline model for placing sensor nodes within the target region, taking into account the coverage requirements and network connectivity. We propose a two-stage centralized control model that ensures cohesive decision making, grouping sensor nodes into protective boxes. This grouping facilitates shared resource utilization, including batteries and bandwidth, while minimizing box number for cost-effectiveness. Noteworthy contributions of this research encompass addressing connectivity and coverage challenges through an offline deployment model in the first stage, and resolving real-time adaptability concerns using an online energy optimization model in the second stage. Emphasis is placed on the energy efficiency, achieved through the sensor consolidation within boxes, minimizing data transmission hops, and considering energy expenditures in sensing, transmitting, and active/sleep modes. Our simulations on an agricultural farmland highlights its practicality, particularly focusing on the sensor placement for measuring soil temperature and humidity. Hardware tests validate the proposed model, incorporating parameters from the real-world implementation to enhance calculation accuracy. This study provides not only theoretical insights but also extends its relevance to smart farming practices, illustrating the potential of WSNs in revolutionizing sustainable agriculture.

## 1. Introduction

Wireless Sensor Networks (WSNs) are systems in which multiple sensors communicate with each other to transmit the data they have collected from a field to a gateway. WSNs can be either homogeneous or heterogeneous, depending on the employed sensor types. A homogeneous sensor network consists of sensors of the same type, while a heterogeneous sensor network comprises sensors of different types. Heterogeneous sensor networks allow the collection of information related to diverse variables from the environment. WSNs, a significant application of Internet of Things (IoT) technology, play a crucial role in facilitating communication among objects, enabling data transfer between them. The aim of these applications is to improve energy efficiency and expedite decision-making processes.

As IoT technology advances, WSNs find utility in various sectors for a multitude of purposes. WSNs are used for various applications, including detecting forest fires, observing wildlife, numerous applications in the health sector, smart farming practices, determining traffic density in smart cities, ensuring border security for military purposes, controlling war zones, and monitoring smart home security systems and devices in smart homes [1].

Sensors have the capability to sense data within their sensing range and transmit the sensed data to other sensors within their communication range [2]. Sensors are devices with limited battery power, low processing capabilities, and limited memory. Therefore, extending the network lifetime in WSNs composed of these sensor nodes is a crucial challenge. To sustain the WSN structure for an extended period, some improvements need to be implemented. The key factors include strategically placing sensors in the right locations and determining the shortest path for the sensor data collected from the field to reach the gateway. The solution to these sub-problems extends the network lifetime.

Efficient sensor placement in a field presents several considerations. Foreknowledge of the specific points to be monitored within the target region is essential. The strategic placement of the sensors must account for the frequency with which scanning areas will be covered by different sensors. Following this distributed placement procedure, the possibility of placing multiple sensors in the same location is explored. In this study, we strategically position the sensors within the sensor nodes distributed across the target region. This packaging strategy is designed to optimize the battery energy consumption by promoting the sharing of a common battery and bandwidth among sensors. Additionally, it contributes to cost reduction in the installation. In this research, we proposed Offline Network Deployment mathematical model to deploy the sensor nodes across a target region. This offline model is used only once before the field monitoring task in order to optimally place the sensors within the target region.

During the transmission of the collected data from the field to the gateway, the data can be lost due to the length of the paths. In the research [3], to prevent the data loss, it is suggested to keep copies of the measured data on neighboring sensor nodes. However, maintaining multiple copies of the same data in the model leads to unnecessary memory usage and, consequently, more battery energy consumption. This study deliberately excludes the scenario of data duplication on the sensors, focusing instead on minimizing the number of hops a data packet undertakes to reach the gateway in striving towards effective energy utilization.

The sensors consume a certain amount of energy when sensing data, transmitting data, and being in the sleep mode [4]. The amount of energy consumed by the sensors is a critical factor directly influencing the network’s lifetime [5]. In this study, bringing sensors together in a box and using a shared battery helps to minimize the amount of energy consumed. In the offline model, accurately placing sensor nodes in the target region also reduces the number of hops, thus decreasing the energy required for the data transmission.

The research study [3] proposes an offline model that calculates the route through which the data will be transmitted to the gateway once. This situation can render the model ineffective in the event of a sensor node failure within the network. This study employs an online mathematical model to ascertain the data flow routes. The node-to-gateway data routes are dynamically adjusted based on the current network state at each time period. Consequently, if a sensor node becomes inactive due to the factors such as a depleted battery or any other reason, the data flow route is automatically recalculated to adapt to the altered network configuration.

Wireless sensor networks can be controlled in centralized or decentralized manners. In the centralized control of a network, the control commands are calculated by a central decision maker, and each network element acts according to the decision made by the center. In the decentralized control of a network, each network element has the potential to make decisions independently of the others [6]. As an example, a sensor node can decide through which sensors it should transmit data to the gateway. In this work, we focus on a centralized WSN control model including offline and online stages. The mathematical models for both stages are executed as a centralized control, and the sensor nodes operate in the network according to the decisions made by the central decision maker.

In this study, a heterogeneous network model is employed, and the sensor nodes are created by grouping sensors. Each sensor node is placed inside a protective box, allowing the node’s battery and bandwidth to be commonly utilized by the sensors within. Decisions are made to minimize the number of boxes when grouping sensors. Using an offline model, the sensor nodes are placed in the field based on the coverage requirements of the target points. In order to secure the network connectivity, there should be at least a minimum number of active nodes within the communication range of a sensor node. Connectivity and coverage problems are addressed through this approach. Taking into account the energy expenditures in sensing, transmitting, active, and sleep modes of the sensors, the sensor node activity schedules are determined under the battery energy constraints. Moreover, our online energy optimization model determines the minimum energy consuming node-to-gateway data flow paths within the bandwidth limits of the sensor nodes.

This study has been simulated on the application of the proposed two-stage mathematical models on an agricultural farmland. The ultimate aim is to deploy the soil temperature and humidity sensors in the farmland in a connected manner, and maximize the network lifetime for smooth information flow from the field. The study is supported by the hardware tests. The parameters are obtained through the hardware implementation, which are then served as the input parameters to our mathematical models. The primary contributions of this research to the literature can be summarized as follows:Deployment of Sensor Nodes Based on Land Coverage Requirements:The study focuses on placing sensor nodes by considering the coverage requirements in agricultural farmland, contributing to the optimization of the field coverage.Utilization of a Heterogeneous Network Model: By adopting a heterogeneous network model, the study groups sensors to create sensor nodes, allowing sensors within a node to leverage the node’s battery and bandwidth.Integration of Offline and Online Mathematical Models: The study employs an offline model to place sensor nodes based on the coverage requirements and an online mathematical model to facilitate data flow from sensor nodes to the gateway in real-time.Comprehensive Examination of Energy Expenditures: Energy consumption, including sensing, transmission, active, and sleep modes, as well as the total battery energy of sensor nodes, is thoroughly examined to effectively manage the energy usage.Resolution of Connectivity and Coverage Issues: The study addresses connectivity and coverage problems by considering the coverage requirements and the minimum number of nodes required around each sensor node during the deployment.Reduction of Hop Count and Enhancement of Data Transmission Efficiency: The proposed model improves energy efficiency by reducing the hop count and ensures efficient data transmission to the gateway, thus minimizing the data loss.

## 2. Literature Review

Numerous studies in the literature have delved into sub-problems related to the focus of this research. A prominent concern in these studies revolves around the impact of sensor placement on energy consumption during data transmission. For instance, this challenge is formulated as a non-concave Boolean optimization problem and proposed a solution method that relaxes both the ’s’ and Boolean constraints to optimize the sensor deployment [7].

Energy efficiency is a key concern in WSNs, and several studies have proposed different approaches to address this challenge. One approach is to strategically position sensors to cover the target area while minimizing the sensor usage [8]. Another approach is to facilitate data exchange among sensors to reduce the volume of data passing through individual sensors [9].

Additionally, WSNs often consist of sensors with diverse technical specifications that jointly perceive a given area while collecting the same data type. This study focuses on the collective placement and grouping of sensors alongside other network elements to maximize the data collection across the entire area over extended periods [10].

In the research [11], a novel approach is proposed to data loss prevention in WSNs. Their approach is based on the idea of grouping sensors in different locations for data transmission, while emphasizing the arrangement of network elements located within the same vicinity. The central theme of this research lies in optimizing data routing to minimize energy consumption, wherein network elements play a crucial role in data transmission. The selection of optimal groupings is determined through an online model. This approach has several advantages over other data loss prevention techniques. First, it is energy-efficient, as it groups sensors together in a way that minimizes energy consumption. Second, it is reliable, as it takes into account the reliability of each sensor when grouping sensors together. Third, it is scalable, as it can be applied to WSNs of any size. This approach has the potential to be used in a variety of WSN applications, such as environmental monitoring, industrial automation, and healthcare.

In addition to the approach proposed in [11], there are a number of other techniques that can be used to mitigate data loss within WSNs. One approach is to use a routing protocol that is specifically designed for WSNs. In the paper [12], a routing approach is proposed based on the MAC (Medium Access Control) protocol that is specifically designed for WSNs. Their approach takes into account the unique characteristics of WSNs, such as the limited energy resources of sensor nodes and the unreliable nature of wireless communication.

Another approach to mitigating data loss is to use error-detection and -correction techniques. In [13], a comparative analysis is conducted between scenarios where sensor transmission times run in parallel and scenarios with shifts in transmission times. They presented a method for detecting such shifts in data transmission. This method can be used to correct errors in data transmission and improve the reliability of WSNs.

Another approach adopted in the literature to prevent data loss is to forward the data to the multipath debris. In the study [14], two separate routes were created using different nodes for data transmission. In order to balance the data load on the network elements and extend the life of the network, prior studies [15,16] identified alternative data paths with convolutional heuristics. Using ant colony optimization, another study [17] avoided routing over high energy-consuming sensors in the network. One study [18] appointed a certain number of sensor group heads on the network to minimize the energy consumed during data transfer between sensors. The sensors that are members of the group select the sensor with the highest remaining energy and the closest one for the next step and transmit their data with multi-step routing per group. The group leader reduces the energy consumption of the network by forwarding the data to the gateway with a pre-calculated route. Identifying multiple paths to reduce data loss increases the amount of energy consumed, while identifying some nodes on a group basis throughout the entire network lifetime concentrates the data load on these nodes. In this work, in order to prevent data loss, our two-stage models find the best path to reach the gateway with minimum number of hops. Since the proposed optimization models detect the least energy-consuming paths in each period, nodes do not need to serve as a group for the entire network lifetime.

In the timing-based TDMA protocol, each network element is assigned a time slot to transmit its data. Network elements sleep to conserve energy during periods when they are not transmitting data. In the contention-based protocol, if the network element does not receive any signal during the random waiting period, it sends a control packet to the neighboring network elements and goes into sleep mode until the next transmission period. Network elements that go into sleep mode increase the likelihood of delays in data transmission. A previous study [11] randomly grouped the sensors to overcome the shortcomings of these two protocols, reducing the number of sensors listening to the network and saving energy. According to this study, the sensor that wants to transmit data to a sensor in a different group will perform this transmission as long as the target group is active. There are different studies on time and contention-based protocols in the literature [19,20,21]. In this work, the online energy optimization model determines the the least energy consuming data transmission paths.

The data routing and scheduling of sensors’ active/sleep periods have been investigated in the literature to minimize energy use in data transmission in the network [22]. In [23], the aim is to provide energy efficiency by grouping active/sleep modes and aimed to transmit data to the gateway as soon as possible. [24] suggested different sleep modes for energy saving: in deep sleep, the sensor is completely off until a certain time, while in light sleep it is ready to wake up. The authors of [10] proposed a mixed integer linear model for different purposes such as shortest data transmission path and maximum data transport. In this model and [25], it has been observed that the sensor that transmits data directly to a gateway has a high amount of data and is the fastest depleted of energy in the network.

In [26], three different heuristic routing approaches are compared: proactive routing, which is determined and not updated, reactive routing, which is updated as needed, and opportunistic routing, where each sensor controls data transmission. When a sensor is de-energized, data sensing/transmission tasks cannot be loaded on the network. According to [27], a data orientation towards energy efficiency is provided by clustering the sensors. The closest study to the proposed work is [28], which detects the same type of data, covers the entire area, and consists of sensors with different technical characteristics that will transmit data to the gateway with the least energy, determining expenses and routes. The main differences of this work to the current study can be listed as follows. (1) Network elements include an actuator and routers in addition to sensors and receivers. (2) Sensors detect different types of data and have different data detection and transmission. In periods, (3) data collected from the field are evaluated; then, commands will be sent to the actuator by the gateway, and the data flow must be determined bidirectionally in order for the gateways to give data confirmation to the network elements and send sleep/active commands. (4) The data will be forwarded by a computer/online method based on gateway recognition. (5) A minimum number of boxes (grouping), so that the network elements can withstand environmental factors, will be placed on the field.

In our preliminary work [3], we have considered a centralized mathematical formulation for a physical layer, MAC layer, and network layer of a WSN for a monitoring mission. In this work, we extend our approach in the following ways: (1) we consider the problem in two separate stages, i.e., offline network deployment and online energy operations models, (2) we propose a novel approach to handle data loss in transmission that can propose alternative data route paths even if some of the network elements fail, (3) we aim to collect different types of sensors in one protective box including a common battery and transmitter. That is, the sensors will share the same limited energy and the transmitter bandwidth.

Based on the observation that large data packets sent at frequent intervals shorten the battery life of the sensors, data compression has been evaluated in the literature to reduce energy use [29,30]. In [31], the amount of energy lost during data transmission is reduced by decreasing the amount of data accepted or transmitted with the proposed compression algorithm. In this work, the main approach to reduce energy use in data transmission will be to transmit data from the shortest distance to the gateway, with the secondary goal being to transmit sensor measurements in small data packets.

Different modules have been used in the literature for the embedded system and mechanical design of sensor networks. On the other hand, [32] provided the communication in the sensor network they created to observe an agricultural field with the XBee protocol at 2.4 GHz radio frequency in the ISM band. Atmega8L micro-controller, XBee RF module, EEPROM memory are located in the sensor nodes to save measurement data and prevent data loss. [33] selected the MSP430F1611, which is known for its ultra-low power consumption, and the nRF24L01 module, which is known for its ultra-low, mid-range and low power consumption, as well as a micro-controller for a low-power sensor node known for its low cost and for working in the ISM band 2.4 GHz frequency as a radio transceiver. The nRF24L01 and the micro-controller are put into sleep mode with the sleep algorithms created during the periods when there is no data flow. However, the hardware and software of the gateways where the data were collected were not examined.

Comparing in detail different sensor network designs in terms of power consumption and data capacity, [34] designed a low-power sensor node. In this design, ultra-low power consumption MSP430F1611 and TI250CC are used as a micro-controller and radio transceiver, respectively (Texas Instruments, Dallas, TX, USA, 2020). While the power requirement of the node board is provided by AAA alkaline batteries, no design is recommended for the gateway. This work will use LoRa operating module, EEPRROM for sensors and actuator, and SD card memory for router and gateways. In the work, separate electronic board designs will be made so that the sensors, actuator, routers and gateways in the network can work with low energy consumption. In the 3D site, the external mechanical design that will protect the sensors against conditions such as high temperature, humidity, dust for the life of the network.

## 3. System Description

WSNs consist of low-cost sensors with limited communication, sensing, and storage capabilities. Today, with the widespread use of technology, WSNs are used in various fields such as collecting health data, scanning military areas, monitoring natural life and environmental conditions, and agriculture [25]. Many agricultural and livestock applications of sensor networks have been used to report the ongoing behavior of animals, to report a possible abnormality, and to obtain more efficiency from animals and plants by measuring the humidity and temperature of the environment [35].

The sensor network discussed in this work basically consists of the network elements such as sensor node, actuator, and gateway as shown in Figure 1. The sensor node contains different types of sensors and a router, shown in Figure 2.

The sensors take measurements in the field at regular intervals and transmit data packets to the routers or sensors with which they can communicate to transmit information to a gateway. Routers are able to communicate with other network elements. However, they are not capable of sensing data in the field. Radio frequency (RF) provides the communication amnong the sensor nodes, routers, and the gateways. The network data are collected in cloud storage and evaluated by a decision maker, and commands are sent to the actuators via gateways by using RF communication. According to these commands, actuators perform functions such as irrigation and lighting.

The target region includes *N* candidate points that are collected in the set N={1,…,N}. A point j∈N is a candidate position to deploy a network element, and also a location that a sensor measurement is requested from. There are *K* types of sensors which are indexed by the set $K_{1}=\left\{1,\ldots ,K\right\}$. There are different types of network nodes: a type *s* node, i.e., a *sensor node*, which includes multiple sensors located at the same point, a type *a* node is an actuator, a type *r* node is a router, and a type *g* node represents a gateway. The node types are collected in the set $K_{2}=\left\{s,a,r,g\right\}$. A sensor (j,k) represents a type $k\in K_{1}$ sensor deployed at point j∈N.

A type *k* sensor can sense a point j∈N if it falls into the sensing range $r_{k}^{s}$, and it can send data to other network elements (sensor, router, gateway) within the communication range $r_{k}^{c}$. As illustrated in Figure 3, the sensor (j,k) can take measurements for the point $p_{1}$ since it is inside the sensing range $r_{k}^{s}$. After processing the collected data of the point $p_{1}$, the sensor (j,k) can transmit the data packet to the sensor $s_{1}$, which is inside the communication range $r_{k}^{c}$. Note that the sensor (j,k) can neither sense the point $p_{2}$, nor can it transmit data to the sensor $s_{2}$ since they are beyond the corresponding operation ranges.

A sensor (j,k) costs $c_{jk}$ monetary units with an initial battery energy of $E_{k}$ mAh. A sensor node is no longer a member of a WSN when its battery is over. On the other hand, a sensor node can save significant amount of energy by going to the sleep mode when it is not sensing or transmitting data, which prolongs its lifetime.

A sensor node (j,k) for $k\in K_{1}$ consumes $e_{k}^{a}$ mAh to stay active without sensing data (Figure 4a), and $e_{k}^{sr}$ mAh to sleep for one period (Figure 4c). In the sleep mode, a sensor cannot measure, receive or transmit data, but it listens to a wake-up signal from a gateway [36]. A type *k* sensor needs $e_{k}^{s}$ mAh to collect and process data in a period. A sensor node (j,k) spends $e_{k}^{ar}$ mAh to receive unit data from other nodes, and $e_{k}^{c}$ mAh to transmit unit data to other nodes (Figure 4b). The amount of data a type *k* sensor can transmit per unit of time is limited by the bandwidth $\beta_{k}$ which is utilized by the sensor’s own data and data from other sensors.

We aim is to deploy gateways to transmit the collected network data to the cloud. It is assumed that the communication range of the gateways is large enough to communicate with all elements in the field. It is also accepted that the battery energy of the gateways is sufficient to supply the entire planning horizon. Due to these superior features of gateways, their costs are higher than sensors and they are found to be in less number in the network. The number of elements that can be used in the network must be within the limited budget *B*.

A sensor network is expected to operate smoothly in the field for a long time. It is considered that the network elements are kept in a protective box in order to withstand environmental conditions such as rain and sun. In this work, our target is to group the network elements using as few boxes as possible.

We also aim to plan the sleep and active mode periods of the sensors in the best way in order to use the limited $E_{k}$ battery energy for the longest time. Another objective is to determine the number and locations of the network elements that can sense the terrain by communicating with each other for the longest time within the constrained *B* budget.

The sub-problems addressed in the network can be listed as follows:Deployment of the network elements: Among the candidate positions, the best sensor nodes locations are determined to monitor the target points with desired quality. The connectivity of the network is supported by the deployment of routers. The gateways should be located to oversee each network element. The decisions should be made within the budget limitation.Sensor activity scheduling: In order to save the battery energy, a sensor can go into the sleep mode, which may harm the coverage requirements and the network connectivity. The energy-optimized activity schedule of the sensor nodes should be determined to obey the connected coverage restrictions during the planning horizon.Data routing: The sensor data should be routed to a gateway at each time period. As the network gets larger, direct communication with a gateway becomes impossible due to the node communication ranges. In that case, the data can follow a multi-hop node-to-gateway transmission route. As the number of hops increases, the transmission failure probability gets higher due to the bandwidth limitations, interference and noise in the environment. This may end up with the data loss which is undesired. Then, data loss-aware energy-optimized data transmission routes should be generated.

## 4. Sensor Network Design

In order to design a WSN, we first introduce mathematical formulations to optimize the network deployment, activity scheduling, and data routing operations in Section 4.1 and implement on a hardware in Section 4.2.

### 4.1. Mathematical Models

We consider the WSN optimization problem at two stages: the offline network deployment (OND) model given in Section 4.1.1 and the online energy optimization (OEO) model explained in Section 4.1.2. The OND model deploys the network agents, such as sensors, actuators, routers, and gateways, within the target region at the beginning of the monitoring mission. OND aims to minimize the deployment cost while ensuring that the deployed sensors are sufficient to monitor the target region within a desired sensing and communication quality. In order to maximize the network lifetime *L*, OND deploys redundant sensor nodes in a WSN.

Secondly, the OEO model considers the agent locations determined by the OND model and aims to determine the minimum energy consuming sleep/active schedule of the sensors while satisfying the connected coverage requirements of the mission. Being an online model, OEO also determines the minimum energy consuming sensor to gateway data transmission routes at each period *t*. Finally, the network lifetime can be determined as the first period *L* at which the OEO model fails to generate a feasible solution. That is the time at which the battery energy of some sensor nodes are depleted and the alive nodes cannot cover the region in a connected manner. We summarize the parameters and decision variables in the OND and OEO models in Table 1.

#### 4.1.1. Offline Network Deployment Model

The OND model deploys the sensor nodes to monitor each target point while forming a connected network with the other sensor nodes and routers with minimum budget usage. In order to prolong the network lifetime and ensure the existence of data transmission routes to a gateway, OND locates redundant sensor nodes in the network.

An actuator is a fixed equipment such as an irrigation valve or a greenhouse light. Hence, we assume that the actuator locations are known and not modeled. That is, the OND model aims to find the optimum locations of sensor nodes for each sensor type, routers and gateways. The binary decision variable $x_{jk}$ indicates if there is a sensor or a router ($k\in K_{1}\cup \left\{r\right\}$) at point *j* or not. Similarly, the binary variable ${\bar{x}}_{j}$ represents the gateway deployment decision at point *j*. If there is a node deployed at point *j*, then the binary variable ${\bar{y}}_{j}$ is one and zero otherwise.

The OND objective function (Equation 1) minimizes the total deployment cost of sensors, routers, and gateways. In particular, the parameter $c_{k}$ represents the cost of a type *k* element, $k\in K_{1}\cup \left\{r,g\right\}$ and ${\bar{c}}_{j}$ is the fixed cost of deploying a node at point j∈N. In order to protect the hardware from harsh environmental conditions, such as rain and high temperature, we consider inserting network elements in a protective box. Hence, ${\bar{c}}_{j}$ can be considered as the fixed box cost and the decision variable ${\bar{y}}_{j}$ aims to minimize the number of utilized nodes in the network.
(1)$$\begin{array}{cc} \min & \;\sum_{j\in N,\;k\in K_{1}\cup \left\{r\right\}}c_{k}x_{jk}+\sum_{j\in N}c_{g}{\bar{x}}_{j}+\sum_{j\in N}{\bar{c}}_{j}{\bar{y}}_{j}\\ s.t. & \end{array}$$
(2)$$\begin{array}{cc} \; & \sum_{j\in N,\;k\in K_{1}\cup \left\{r\right\}}c_{k}x_{jk}+\sum_{j\in N}c_{g}{\bar{x}}_{j}+\sum_{j\in N}{\bar{c}}_{j}{\bar{y}}_{j}\leq B \end{array}$$
(3)$$\begin{array}{cc} \; & \;\;\sum_{j\in N}a_{jki}x_{jk}\geq {\bar{f}}_{ik}\;\;\;\;\;i\in N,\;k\in K_{1} \end{array}$$
(4)$$\begin{array}{cc} \; & \;\;\sum_{j\in N}b_{jgi}{\bar{x}}_{j}\geq 1\;\;\;\;\;\;i\in N \end{array}$$
(5)$$\begin{array}{cc} \; & \;\sum_{\begin{array}{c} j\in N,\;k\in K_{1}\cup \left\{r\right\}\\ \left(j,k)\neq (i,l\right) \end{array}}b_{ilj}x_{jk}\geq \alpha x_{il}\;\;i\in N,\;l\in K_{1}\cup \left\{r\right\} \end{array}$$
(6)$$\begin{array}{cc} \; & \;\;\;x_{jk},\;{\bar{x}}_{j},\;{\bar{y}}_{j}\in \left\{0,1\right\}\;\;\;j\in N,\;k\in K_{1}\cup \left\{r\right\} \end{array}$$

Constraint (Equation 2) limits the total deployment cost with the available budget *B*. The binary parameter $a_{jki}$ indicates whether a sensor (j,k) can sense point *i* or not. Constraints (Equation 3) guarantee that the deployed sensors can satisfy the coverage requirement ${\bar{f}}_{ik}$ of a point *i* by a type *k* sensor. Here, the parameter ${\bar{f}}_{ik}$ picked is greater than the true coverage requirement of a point *i*. This allows the deployment of redundant sensor nodes in the network within the available budget *B*, which can help to maximize the network lifetime *L*.

The binary parameter $b_{ilj}$ indicates whether a sensor (i,l) can communicate with a node at point *j* or not. The gateways should communicate with each network element in order to command them for the active/sleep modes and routing rules. Constraints (Equation 4) deploy the gateways to communicate with all target points. In order to secure the network connectivity, we assume that each sensor or router node should communicate with at least α different nodes as dictated by Constraints (Equation 5). Finally, Constraints (Equation 6) set the binary restrictions on the decision variables.

#### 4.1.2. Online Energy Optimization Model

The OEO model determines the active sensor nodes and generates the minimum energy consuming data transmission routes from each node to a gateway at each time period *t* by considering the bandwidth limitations of the nodes. OEO activates the sensor nodes without redundancy, which are sufficient to satisfy the coverage requirements of the target points. The efficient activity scheduling and energy-aware data routing of the OEO model allows the maximization of the network lifetime *L*.

In the OEO model, the binary decision variable $z_{jkt}$ is one if a sensor node (j,k) is active at period *t* and zero otherwise. The continuous decision variable $y_{iljkt}$ shows the amount of data sent from a sensor (i,l) to sensor (j,k) at period *t*. The energy consumption of a node (j,k) in a period *t* is decided by the continuous variable $E_{jkt}$, which can be given in Equation (Equation 7).
(7)$$E_{jkt}=e_{k}^{sr}\left(1-z_{jkt}\right)+e_{k}^{a}z_{jkt}+e_{k}^{ar}\sum_{\left(i,l\right)\in N\times K_{2}}y_{iljkt}+e_{k}^{c}\sum_{\left(i,l\right)\in N\times K_{2}}y_{jkilt}+\sum_{j\in N}e_{k^{\prime }}^{s}z_{jkt}x_{jk}$$

The third term in Equation (Equation 7) is the inflow energy, fourth term is the outflow energy, and fifth term is the sensing and processing of the sensor (j,k).

The energy consumption $E_{jrt}$ of a router node (j,r) in a period *t* depends on the sleep and active modes, total inflow and total outflow as shown in Equation (Equation 8).
(8)$$E_{jrt}=e_{r}^{sr}\left(1-z_{jrt}\right)+e_{r}^{a}z_{jrt}+e_{r}^{ar}\sum_{\left(i,l\right)\in N\times K_{2}}y_{iljrt}+e_{r}^{c}\sum_{\left(i,l\right)\in N\times K_{2}}y_{jrilt}$$

Equation (Equation 9) gives the energy consumption $E_{jat}$ of an actuator node (j,a) in a period *t* as the summation of sleep, active and transmit energy consumption. We assume that an actuator does not operate as a relay node, but only transmits its own data.
(9)$$E_{jat}=e_{a}^{sr}\left(1-z_{jat}\right)+e_{a}^{a}z_{jat}+e_{a}^{c}\sum_{\left(i,l\right)\in N\times K_{2}}y_{jailt}$$

We assume that a gateway (j,g) is active throughout the mission. Then, the energy consumption $E_{jgt}$ of a gateway (j,g) in a period *t* includes the active and inflow energy consumption as in Equation (Equation 10).
(10)$$E_{jgt}=e_{g}^{a}+e_{g}^{ar}\sum_{\left(i,l\right)\in N\times K_{2}}y_{iljgt}$$

The OEO objective (Equation 12) aims to minimize the total energy consumption at a period *t*. Being an online model, OEO takes the remaining energy $E_{jkt}^{rem}$ of a node (j,k) as an input to optimize the energy consumption $E_{jkt}$ in the current period *t*. Equation (Equation 11) calculates the remaining energy of a node $\left(j,k\right)\in N\times K_{2}$ at period *t* iteratively when the energy consumption $E_{jk(t-1)}$ is given.
(11)$$E_{jkt}^{rem}=E_{jk(t-1)}^{rem}-E_{jk(t-1)}$$

Constraints (Equation 13) set an upper bound $E_{jkt}^{rem}$ on the energy consumption of a node (j,k). Constraints (Equation 14) allow to activate a node if it has been deployed by the OND model, i.e., $x_{jk}=1$. Constraints (Equation 15) activate sensor nodes to satisfy the coverage requirement of a point i∈N. According to Constraints (Equation 16), there should be at least α many active nodes in the neighborhood of an active node (i,l) for connectivity.
(12)$$\begin{array}{ccc} \min & \;\sum_{\left(j,k\right)\in N\times K_{2}}E_{jkt}\\ s.t. & \; & \end{array}$$
(13)$$\begin{array}{cc} \; & E_{jkt}\leq E_{jkt}^{rem}\;\left(j,k\right)\in N\times K_{2} \end{array}$$
(14)$$\begin{array}{cc} \; & z_{jkt}\leq x_{jk}\;j\in N,\;k\in K_{1}\cup \left\{r\right\} \end{array}$$
(15)$$\begin{array}{cc} \; & \sum_{j\in N}a_{jki}z_{jkt}\geq f_{ik}\;i\in N,\;k\in K_{1} \end{array}$$
(16)$$\begin{array}{cc} \; & \sum_{\left(j,k\right)\in N\times K_{2}}b_{jki}z_{jkt}\geq \alpha z_{ilt}\;\left(i,l\right)\in N\times K_{2} \end{array}$$
(17)$$\begin{array}{cc} \; & y_{iljkt}\leq \bar{M}b_{ilj}z_{ilt}\;\left(i,l\right),\left(j,k\right)\in N\times K_{2} \end{array}$$
(18)$$\begin{array}{cc} \; & y_{iljkt}\leq \bar{M}b_{ilj}z_{jkt}\;\left(i,l\right),\left(j,k\right)\in N\times K_{2} \end{array}$$
(19)$$\begin{array}{cc} \; & y_{ililt}=0\;\left(i,l\right)\in N\times K_{2} \end{array}$$
(20)$$\begin{array}{cc} \; & \sum_{\left(i,l\right)\in N\times K_{2}}y_{iljkt}+h_{jk}z_{jkt}=\sum_{\left(i,l\right)\in N\times K_{2}}y_{jkilt}\;j\in N \end{array}$$
(21)$$\begin{array}{cc} \; & \sum_{\left(i,l\right)\in N\times K_{2}}y_{iljkt}=\sum_{\left(i,l\right)\in N\times K_{2}}y_{jkilt}\;\left(j,k\right)\in N\times \left\{r,a,g\right\} \end{array}$$
(22)$$\begin{array}{cc} \; & \sum_{\left(i,l\right)\in N\times K_{2}}y_{iljkt}+h_{jk}z_{jkt}+\sum_{\left(i,l\right)\in N\times K_{2}}y_{jkilt}\leq \beta_{s}\;j\in N \end{array}$$
(23)$$\begin{array}{cc} \; & \sum_{\left(i,l\right)\in N\times K_{2}}y_{iljkt}+\sum_{\left(i,l\right)\in N\times K_{2}}y_{jkilt}\leq \beta_{k}\;\left(j,k\right)\in N\times \left\{r,a,g\right\} \end{array}$$
(24)$$\begin{array}{cc} \; & y_{iljkt},E_{jkt}\geq 0\;\left(i,l\right),\left(j,k\right)\in N\times K_{2} \end{array}$$
(25)$$\begin{array}{cc} \; & z_{jkt}\in \left\{0,1\right\}\;j\in N,\;k\in K_{1}\cup \left\{r\right\} \end{array}$$

Constraints (Equation 17)–(Equation 23) regulate the inflow and outflow of a node. In order to send a data packet from a node (i,l) to a node (j,k), the point *j* should be within the communication range of the node (i,l), i.e., $b_{ilj}=1$, and the sender and receiver nodes should be active at the same time, i.e., $z_{ilt}=z_{jkt}=1$. Constraints (Equation 17) and (Equation 18) satisfy these requirements by setting a sufficiently large upper bound $\bar{M}$ on the flow amount $y_{iljkt}$. One can pick $\bar{M}$ as the multiplication of the total number of sensors with the maximum data packet size.

A node (i,l) cannot send data to itself as in Constraints (Equation 19). A sensor node (j,k) generates $h_{jk}$ unit of data in an active period. Constraints (Equation 20) set the balance between the inflow, generated data and the outflow for an active sensor node (j,k). A type k∈{r,a,g} node does not sense data. Hence, the inflow should be equal to the outflow as in Constraints (Equation 21). Note that, the energy consumption increases as the data transmission route gets longer. That is, the OEO model generates min-hop data route which decreases the probability of transmission failure among the nodes. As a result, the flow constraints avoid data loss by ensuring the existence of an energy efficient and min-hop data transmission route from each active sensor node to a gateway.

A node (j,k) has a bandwidth $\beta_{k}$ limiting the data flow on the node. The bandwidth $\beta_{s}$ of a sensor node is utilized by the inflow from the other nodes, sensor’s own data and the outflow to the other nodes as in Constraints (Equation 22). For a type k∈{r,a,g} node, the bandwidth $\beta_{k}$ is utilized only by the inflow and outflow data as in Constraints (Equation 23). Constraints (Equation 24) ensure the non-negativity of the flow variable $y_{iljkt}$ and energy consumption $E_{jkt}$. Finally, Constraints (Equation 25) set the binary restriction on the activity schedule variable $z_{jkt}$.

### 4.2. Hardware Implementation

To evaluate the performance of the OND and OEO models discussed earlier, an embedded system emphasizing low power consumption has developed. The design process, guided by mathematical modeling principles, yielded two primary components: gateway and sensor nodes. The system comprises two distinct sensor nodes, each equipped with different sensor types, while a singular gateway oversees their operational activities. The sensor nodes were purposefully diversified to accommodate various sensing functions. Communication between the gateway and sensor nodes is facilitated by a wireless communication protocol. This designed system allows for a comprehensive examination of its functionality, efficiency, and communication protocols, particularly focusing on the wireless connectivity that enables seamless data transfer between the gateway and sensor nodes.

#### Gateway Design

To regulate the operational modes of the sensor nodes, a gateway has been designed and implemented. The hardware configuration of the gateway incorporates an Atmega 328p micro-controller coupled with an E32 LoRa module, serving for wireless communication. The micro-controller possesses the intrinsic capability to interface seamlessly with a diverse array of sensors, encompassing functionalities such as humidity, temperature, and pressure, among others.

Based on the Figure 5, the connections between LoRa and the micro-controller have been established. The gateway, in conjunction with the software, monitors the operational status of the sensor nodes and facilitates the presentation of sensor measurements to users. To ensure low power consumption in the model, the sensor nodes do not continuously measure, as explained in Section 4.3. Instead, the gateway intermittently signals the sensor nodes, prompting them to take measurements and transmit the data back to the gateway. In this system design, two sensor nodes and a gateway communicate at frequencies of 432 MHz and 433 MHz over the second and third channels, respectively. Using the developed software, the gateway sends a signal to the sensor nodes in sleep mode, instructing them to wake up at specific intervals (initially set at every 30 min for the first test phase). The message transmitted contains the intended operation, such as taking a measurement or configuring the node for type-1, type-2, or gateway functions. Subsequently, after receiving measurements from the sensor nodes, the gateway transmits this data to the user.

### 4.3. Sensor Node Design

In this study, the SHT3X air temperature and humidity sensor were utilized in the sensor nodes. Alongside this specified sensor, the Atmega 328p served as the microprocessor for the sensor node. Similar to the gateway, the same LoRa module facilitated wireless communication. The sensor node operated with low power consumption, necessitating the development of different software tailored to the sensor node’s functionality.

The block diagram in Figure 6 illustrates the proposed circuit for the adaptive network. The first sensor node utilized the SHT3X sensor for monitoring air temperature and humidity, making it suitable for outdoor applications. The Atmega 328p acted as the micro-controller, and a LoRa module served for wireless communication. Power was supplied by four AA batteries, providing approximately 5.5 volts. The sensor node was securely housed in a waterproof IP65 box designed for outdoor environments. This sensor node communicated with the gateway on the second LoRa channel at around 432 MHz frequency. Specialized software was developed to achieve low power consumption in the sensor nodes. The sensor node operated initially in sleep mode, waiting for a wake-up signal from the gateway. Upon receiving the wake-up signal, the system became active, measurements were taken, data was sent to the gateway, and then the sensor node returned to the sleep mode, awaiting the next wake-up signal. This operation minimized power consumption. To ensure access to measured data under any conditions and prevent data loss during power outages, a memory system was implemented using EEPROM (Electrically Erasable Programmable Read-Only Memory). The EEPROM library facilitated reading and writing bytes. The Atmega 328p micro-controller had 1024 bytes of EEPROM memory, enabling effective recording of measurements. With the developed algorithm for this memory system, the measurements for 4 bytes are compressed to 1 byte. Now from memory, one byte is used for measuring temperature or humidity, and two bytes are used for the index value. In other words, a total of four bytes of memory are spent for one reading. A total of 192 bytes of memory are used in a twenty-four-hour measurement performed at intervals of thirty minutes. Thus, approximately five days of data can be stored in this memory. In the same way, a similar algorithm has been applied in recording humidity values. Thanks to this method, the memory recording process is designed to be most effective by compressing the data.

## 5. Simulation Results

In this section, we conduct experiments to test our hardware design and estimate the system parameters in Section 5.1, and utilize our mathematical models to implement the hardware on a real-size agricultural field in Section 5.2.

### 5.1. Hardware Results

The current consumption of the circuit during operation is shown in detail in Table 2. The sensor node consumes 6 mA when in sleep mode, 31 mA when sensing and transmitting, 89 mA only when transmitting, and 21 mA only when receiving data.

Figure 7 shows that the simulation results for the sensor’s varying current consumption levels over time in different operating modes with a fixed 5 Volt input. As shown, the sensor switches between sleep, receive, measure, and transmit modes, each characterized by unique current profiles. During the sleep mode, the sensors operate in an energy-efficient standby mode, drawing minimal power. When switching to the receiving mode, the current noticeably increases, indicating the power requirements related to obtaining the signal. In the measurement mode, dedicated to measuring the humidity and temperature values, an average current level is displayed. Then, in the transmission mode, current consumption peaks, reflecting the increased energy demands during data transmission. These results highlight the importance of adjusting power management at each stage of energy efficiency.

In this research, an innovative model has been introduced to address the specific requirements of an adaptive sensor network tailored for low-power Internet of Things (IoT) applications. The proposed architecture focuses on the integration of configurable sensor nodes, strategically designed to achieve an unparalleled level of power efficiency across various application scenarios. To achieve this goal, the model parameters and formulations essential for the seamless functioning of the proposed system were meticulously derived, and these theoretical aspects were subsequently translated into practicality by implementing them at the hardware level.

Taking a step further, the research endeavors included rigorous field testing of the implemented network. Through a series of comprehensive experiments and real-world simulations, empirical evidence was gathered to substantiate the efficacy and viability of the proposed method. In Figure 8 and Figure 9, the hardware implemented in the field is showcased. The results obtained from extensive testing unequivocally affirm that the adaptive sensor network, with its configurable nodes and optimized power efficiency, stands as a promising solution for low-power IoT applications. This study contributes not only to the theoretical foundations of adaptive sensor networks but also provides practical insights that pave the way for the advancement and implementation of energy-efficient solutions in the burgeoning field of IoT technologies.

In the figures, a specific 10-min interval on 18 December 2023 was selected for detailed examination of both temperature and humidity data. The time-stamped temperature and humidity values were extracted and plotted to visualize their variations within this specified timeframe. Figure 10 and Figure 11 present the dynamic changes in both temperature and humidity at regular intervals during this 10 min period. The *x*-axis represents the timestamp, while the *y*-axis corresponds to the temperature (in degrees Celsius) and humidity values, respectively. This comprehensive analysis provides insights into the concurrent fluctuations of temperature and humidity, offering a nuanced understanding of their interplay within this short yet significant temporal window. By examining both parameters, a more holistic view of the environmental conditions during this specific time segment is achieved.

### 5.2. Mathematical Model Results

In this section, we provide the details of a case study to illustrate our OND and OEO models. A sample field is determined with the dimensions 200×250 m. In Figure 12, the eight yellow dots indicate the target points to take sensor measurements and also the candidate positions to deploy network elements, i.e., sensor node, router, and gateway.

Two different sensor types are used in the model, temperature and humidity sensor. Their communication radii are equal to each other and 100 m and their sensing radii are 30 m. Considering the candidate points and the sensing radii of the sensors, each sensor can only detect the point where it is placed. Therefore, the parameter $a_{jkj^{\prime }}$ is calculated as shown in Equation (Equation 26).
(26)$$a_{jkj^{\prime }}=\left\{\begin{array}{cc} 1 & \mathrm{if}\;j=j^{\prime }\\ 0 & \mathrm{if}\;j\neq j^{\prime } \end{array}\right.$$

Since the communication radius of a router used in the model is 500 m, it can communicate with any of the candidate locations in the target region. That is, for a router, we have the parameter $b_{jrj^{\prime }}=1$ for any $j,j^{\prime }\in N$. For a sensor of type k∈{1,2}, the parameter $b_{jkj^{\prime }}$ values are the same due to the identical communication ranges and given as in Equation (Equation 27).
(27)$$b_{jkj^{\prime }}=\left\{\begin{array}{cc} 1 & \mathrm{if}\;j=1\;\mathrm{and}\;j^{\prime }\in \left\{1,2,3\right\}\\ 1 & \mathrm{if}\;j=2\;\mathrm{and}\;j^{\prime }\in \left\{1,2,3,4\right\}\\ 1 & \mathrm{if}\;j=3\;\mathrm{and}\;j^{\prime }\in \left\{1,2,3,4,7\right\}\\ 1 & \mathrm{if}\;j=4\;\mathrm{and}\;j^{\prime }\in \left\{2,3,4\right\}\\ 1 & \mathrm{if}\;j=5\;\mathrm{and}\;j^{\prime }\in \left\{5,8\right\}\\ 1 & \mathrm{if}\;j=6\;\mathrm{and}\;j^{\prime }\in \left\{6,7\right\}\\ 1 & \mathrm{if}\;j=7\;\mathrm{and}\;j^{\prime }\in \left\{3,6,7\right\}\\ 1 & \mathrm{if}\;j=8\;\mathrm{and}\;j^{\prime }\in \left\{5,8\right\}\\ 0 & \mathrm{otherwise} \end{array}\right.$$

In our models, we pick the coverage requirement ${\bar{f}}_{ik}$ of a point *j* as shown in Equation (Equation 28). For connectivity quality, an active node should communicate with at least α=1 many active nodes. The total available budget is *B* = 10,000 TL. The cost of each network element is summarized in Table 3.
(28)$${\bar{f}}_{ik}=\left\{\begin{array}{cc} 0 & \mathrm{if}\;i=8\\ 1 & \mathrm{otherwise} \end{array}\right.$$

For the introduced instance, the OND model generates the solution in Figure 13. According to the results, there is a gateway deployed at point 8. As we mentioned before, the actuator location is given and fixed. In our case, there are two irrigation valves in the field as depicted in Figure 13. Each of the first seven candidate points houses a sensor node including the temperature and humidity sensors.

Moreover, the OND model deployed a router at points 3 and 7 to achieve the connectivity requirements. For instance, the router at point 3 can collect the data of the neighboring sensor nodes and transmit to the gateway energy efficiently. That is, the sensor node at point 3 may not consume energy to route the neighbor node’s data and continue with the sensing operations for a longer time. The gateway at point 8 is out of the communication range of the sensor node at point 7. Then, the OND model locates a router at point 7 to link the sensor node with the gateway.

Equations (29), (30), and (31) summarize the optimal values of the variables $x_{jk}$, ${\bar{x}}_{j}$, and ${\bar{y}}_{j}$ found by the OND model, respectively. Overall, there are seven temperature sensors, seven humidity sensors, two routers, and one gateway deployed with a total cost of 9525 TL.
(29)$$\begin{array}{cc} x_{jk} & =\left\{\begin{array}{cc} 1 & \mathrm{for}\;j\in \{1,2,3,4,5,6,7\}\;\mathrm{and}\;\forall k\\ 0 & \mathrm{for}\;j=8\;\mathrm{and}\;\forall k \end{array}\right. \end{array}$$
(30)$$\begin{array}{cc} {\bar{x}}_{j} & =\left\{\begin{array}{cc} 1 & \mathrm{for}\;j=8\\ 0 & \mathrm{for}\;j\in \{1,2,3,4,5,6,7\}\; \end{array}\right. \end{array}$$
(31)$$\begin{array}{cc} {\bar{y}}_{j} & =\left\{\begin{array}{cc} 1 & \mathrm{for}\;j\in \{1,2,3,4,5,6,7\}\\ 0 & \mathrm{for}\;j=8\; \end{array}\right. \end{array}$$

In the second stage of the WSN operations control, the OEO model determines an activity schedule for each of the network elements and constructs the energy-efficient node-to-gateway data transmission routes. Figure 14 demonstrates the one-period solution of the OEO model based on the network deployment decision of the OND model. According to the solution, each sensor node continuously operates in the active mode until its battery is over. Figure 14 visualizes the node-to-gateway data routes with the labels of flow amounts on the arcs.

As mentioned in Section 4.2, an active sensor generates a data packet of size four bytes, i.e., $h_{jk}=4$ for each period. Based on the energy measurements of hardware simulations in Section 5.1, the 4xAA batteries provide 3000 mAh initial energy for each node ($E_{j}=3000$ for j∈N).

In our simulations, we pick identical coverage requirements ${\bar{f}}_{ik}$ and $f_{ik}$ for the OND and OEO models, respectively. We assume that the bandwidth of each node is sufficient to transmit its own data together with the network flow. OEO is an online model which should be solved repeatedly at each time period. The iterations terminate when there is no feasible solution for the OEO model. Then, the final time period is the network lifetime *L*.

In our instance, the OEO model terminates when node 3 fails to satisfy Constraints (13), which limits the demanded energy with the remaining node energy. The hardware simulations reported in Section 5.1 indicate that a time period is 30 min, and a node data receives data for six seconds, and transmits for two seconds. A sensor node gathers data and processes within three seconds. A network element remains in the sleep mode unless it receives, senses, or transmits during a period.

We assume that the deployed actuators and the gateway have sufficient battery energy to stay active throughout the mission lifetime, i.e., $z_{jat}=1$ and $z_{jgt}=1$. The OEO model reports the optimal schedule for all routers and sensor nodes is to keep the active mode, i.e., $z_{jkt}=1$ for $k\in K_{1}\cup \left\{r\right\}$ for each period *t*. Equation (Equation 32) shows the optimal flow amounts and Equation (Equation 33) is the optimal node energy consumption in a period.
(32)$$y_{iljkt}=\left\{\begin{array}{cc} 4 & \mathrm{for}\;(i,l)\in \{(1,1),(1,2)\}\;\mathrm{and}\;(j,k)\in \{(3,r)\}\;\mathrm{and}\;\forall t\\ 4 & \mathrm{for}\;(i,l)\in \{(2,1),(2,2)\}\;\mathrm{and}\;(j,k)\in \{(3,r)\}\;\mathrm{and}\;\forall t\\ 4 & \mathrm{for}\;(i,l)\in \{(3,1),(3,2)\}\;\mathrm{and}\;(j,k)\in \{(3,r)\}\;\mathrm{and}\;\forall t\\ 32 & \mathrm{for}\;(i,l)\in \{(3,r)\}\;\mathrm{and}\;(j,k)\in \{(8,g)\}\;\mathrm{and}\;\forall t\\ 4 & \mathrm{for}\;(i,l)\in \{(4,1),(4,2)\}\;\mathrm{and}\;(j,k)\in \{(3,r)\}\;\mathrm{and}\;\forall t\\ 4 & \mathrm{for}\;(i,l)\in \{(5,1),(5,2)\}\;\mathrm{and}\;(j,k)\in \{(8,g)\}\;\mathrm{and}\;\forall t\\ 4 & \mathrm{for}\;(i,l)\in \{(6,1),(6,2)\}\;\mathrm{and}\;(j,k)\in \{(7,r)\}\;\mathrm{and}\;\forall t\\ 4 & \mathrm{for}\;(i,l)\in \{(7,1),(7,2)\}\;\mathrm{and}\;(j,k)\in \{(7,r)\}\;\mathrm{and}\;\forall t\\ 16 & \mathrm{for}\;(i,l)\in \{(7,r)\}\;\mathrm{and}\;(j,k)\in \{(8,g)\}\;\mathrm{and}\;\forall t\\ 0 & \mathrm{otherwise} \end{array}\right.$$
(33)$$E_{jkt}=\left\{\begin{array}{cc} 0.08 & \mathrm{for}\;(j,k)\in \{(1,1),(1,2)\}\;\mathrm{and}\;\forall t\\ 0.08 & \mathrm{for}\;(j,k)\in \{(2,1),(2,2)\}\;\mathrm{and}\;\forall t\\ 0.08 & \mathrm{for}\;(j,k)\in \{(3,1),(3,2)\}\;\mathrm{and}\;\forall t\\ 0.67 & \mathrm{for}\;(j,k)\in \{(3,r)\}\;\mathrm{and}\;\forall t\\ 0.08 & \mathrm{for}\;((j,k)\in \{(4,1),(4,2)\}\;\mathrm{and}\;\forall t\\ 0.08 & \mathrm{for}\;(j,k)\in \{(5,1),(5,2)\}\;\mathrm{and}\;\forall t\\ 0.08 & \mathrm{for}\;(j,k)\in \{(6,1),(6,2)\}\;\mathrm{and}\;\forall t\\ 0.08 & \mathrm{for}\;(j,k)\in \{(7,1),(7,2)\}\;\mathrm{and}\;\forall t\\ 0.34 & \mathrm{for}\;(j,k)\in \{(7,r)\}\;\mathrm{and}\;\forall t\\ 0.54 & \mathrm{for}\;(j,k)\in \{(8,r)\}\;\mathrm{and}\;\forall t \end{array}\right.$$

According to the OEO model results, the network survives for a total of 3592 periods (L=3592). This is equivalent to say that the network stays operational for 74 days and approximately 2 months. Figure 15 gives the amount of months that each node can operate according to the OEO model results. The dotted red line shows which node has minimum lifetime in the model. We observe that node 3 consumes the highest energy per period with the shortest lifetime, hence dictates the network lifetime. Figure 14 supports this finding since we observe that node 3 collects the data of the nodes 1, 2, 4, and transmits to the gateway after adding its own data. This corresponds to a data packet of size 384 bytes, which is the maximum flow in the network. The node lifetimes in Figure 15 point out that one can improve the network lifetime by deploying a new router at node 3 or increasing the initial battery energy of the existing sensor node 3.

To sum up, the OND model deployed the nodes in a connected way and the OEO model generates minimum energy consuming data paths by considering the flow balance constraints, which also minimizes the number of hops on the routes. A data transmission attempt may fail due to many reasons and can result in a data loss. Hence, as the OEO model minimizes the number of hops towards a gateway, it diminishes the data loss probability as well. In the end, our two stage WSN control strategy assists the network in pursuing the monitoring mission for maximized duration with minimized data loss.

## 6. Conclusions

In this work, a two-stage mathematical model and its hardware implementation are proposed for the problem of distributing sensors on an agricultural land and collecting data from field in order to increase crop yield. IoT-based smart agriculture technology will facilitate the farmer’s control over the field, minimize the labor force and contribute to the production.

This work focuses on the implementation of wireless sensor networks in agriculture. To efficiently utilize the limited energy resources, optimization models have been developed. The objective is to minimize network deployment costs using the Offline Network Deployment Model (see Section 4.1.1) and optimize the network operation process through the Online Energy Optimization Model (see Section 4.1.2).

A hardware design for a sensor node and a gateway is proposed in order to facilitate the optimal solutions of our mathematical formulations (see Section 4.2). The designed hardware is utilized to obtain the energy measurement parameters which are then used as the mathematical model simulation inputs (see Section 5.1). According to the obtained results, the designed IoT network costs 9525 TL and operated for 74 days.

Our mathematical models generate the optimal locations of the network elements, activity schedules and data transmission routes centrally. That is, there is a central decision maker that has full access to the network element data at any time period and decide optimally by solving the models. However, a WSN can include many network elements which increases the number of variables in the mathematical models as well as the computational time. Moreover, the failure of the central decision maker hinders the regular operation of the network. Another drawback with a centralized model is that obtaining the current state of the entire network is not an easy task due to the limited communication ranges and different clock times of the network elements. From these perspectives, as a future research track, one can consider to design a distributed network model in which each network element makes its own decisions by utilizing the local information. This type of network control will be scalable and robust to the network element failures. On the other hand, such a distributed algorithm can only generate local optimal solutions in contrast to the global optimum solution of a centralized algorithm as in this research.

## Figures and Tables

**Figure 1 sensors-24-01457-f001:**
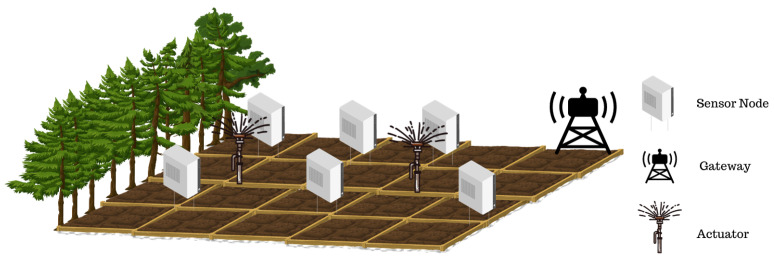
Smart farm area.

**Figure 2 sensors-24-01457-f002:**
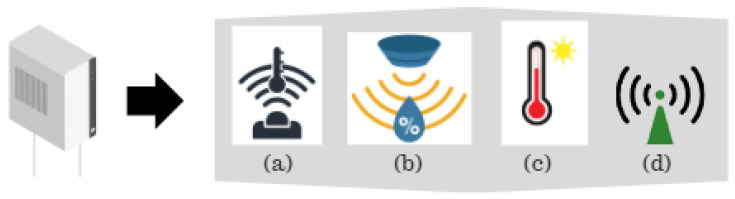
Contents of a sensor node. (**a**) shows temperature sensor, (**b**) shows humidity sensor, (**c**) shows sunbathing time sensor, and (**d**) shows a router.

**Figure 3 sensors-24-01457-f003:**
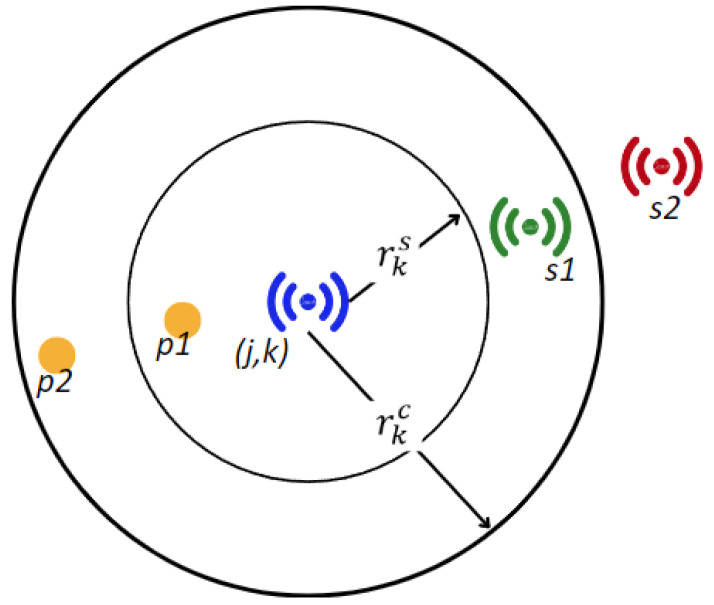
Sensing and communication ranges of a sensor.

**Figure 4 sensors-24-01457-f004:**
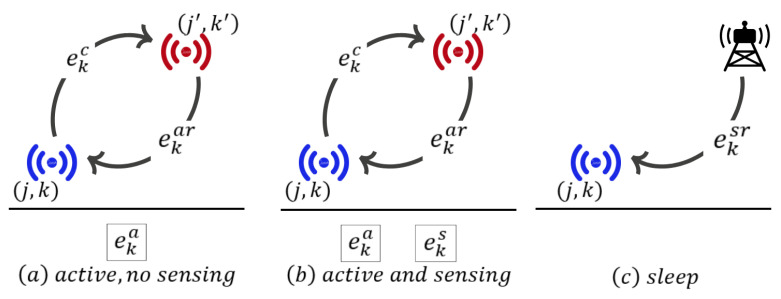
Energy parameters of a sensor.

**Figure 5 sensors-24-01457-f005:**
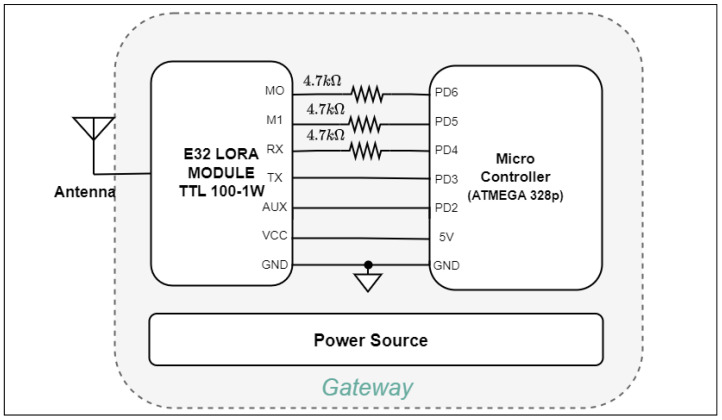
Block diagram of the gateway design.

**Figure 6 sensors-24-01457-f006:**
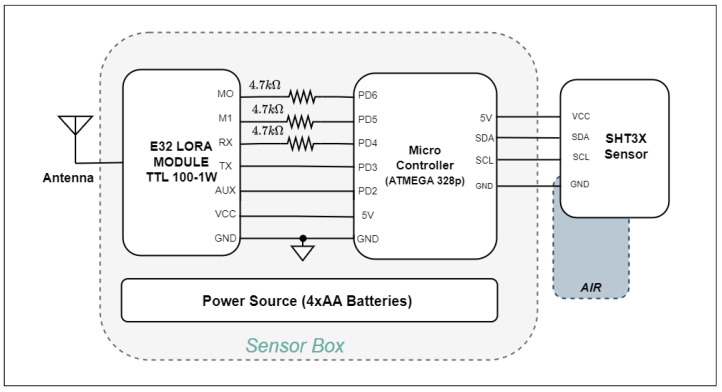
Block diagram of the embedded system designed for the implementation of the proposed adaptive network.

**Figure 7 sensors-24-01457-f007:**
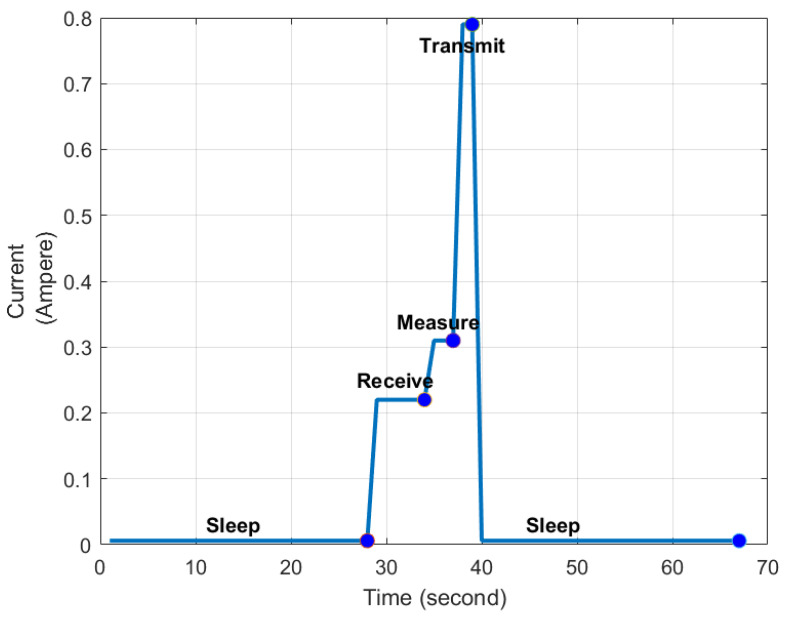
The trace of the current consumption during operation.

**Figure 8 sensors-24-01457-f008:**
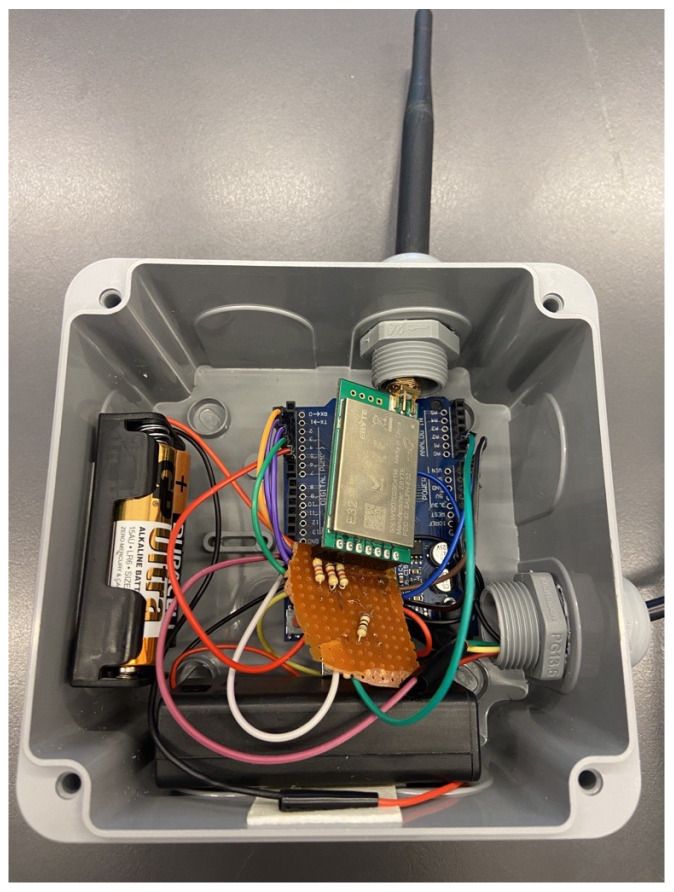
The implemented hardware placed on the field.

**Figure 9 sensors-24-01457-f009:**
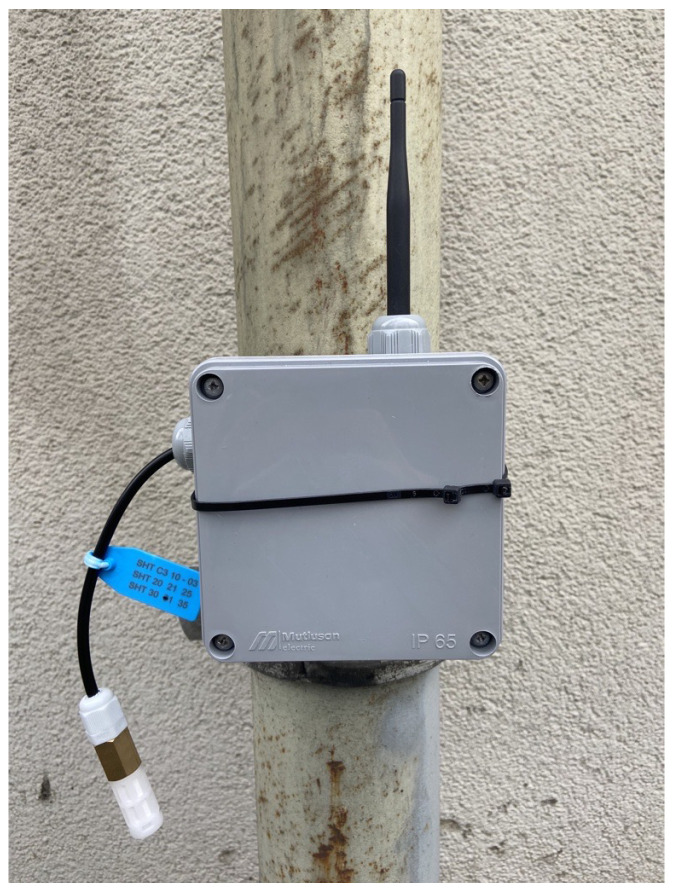
The implementation of a sensor node.

**Figure 10 sensors-24-01457-f010:**
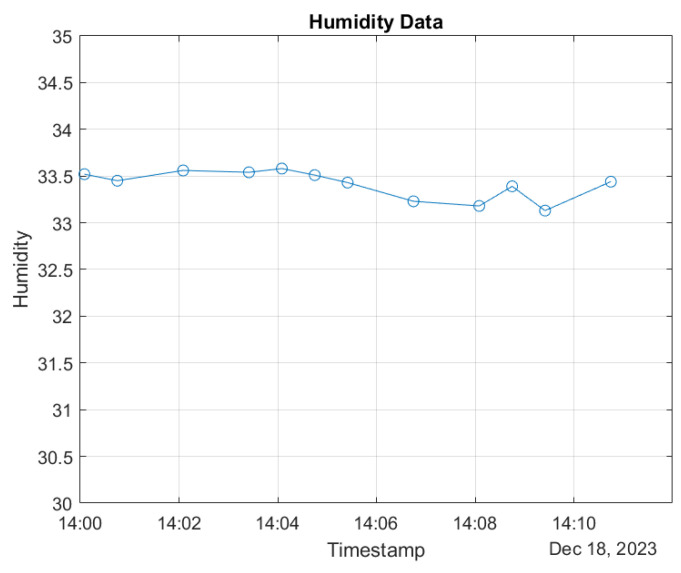
Time-stamped humidity data.

**Figure 11 sensors-24-01457-f011:**
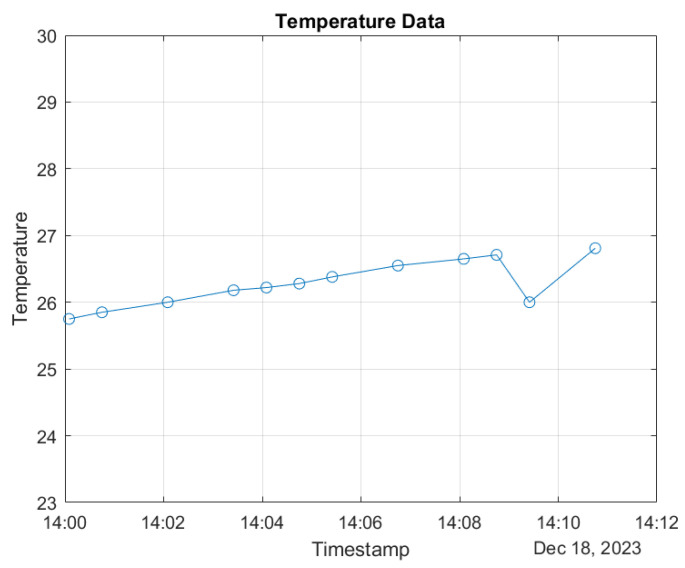
Time-stamped temperature data.

**Figure 12 sensors-24-01457-f012:**
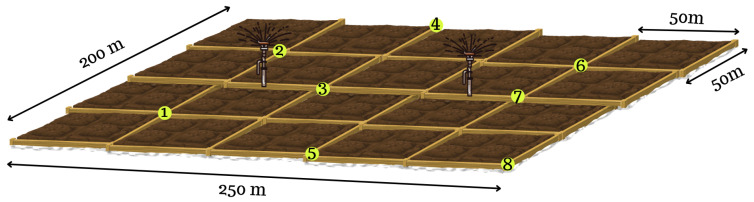
Sample field.

**Figure 13 sensors-24-01457-f013:**
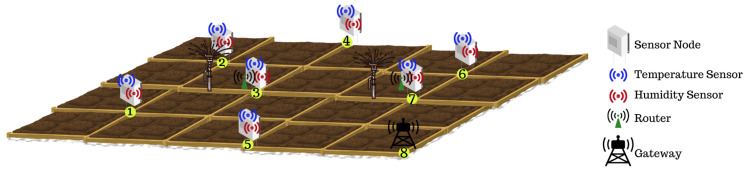
Solution of the OND model.

**Figure 14 sensors-24-01457-f014:**
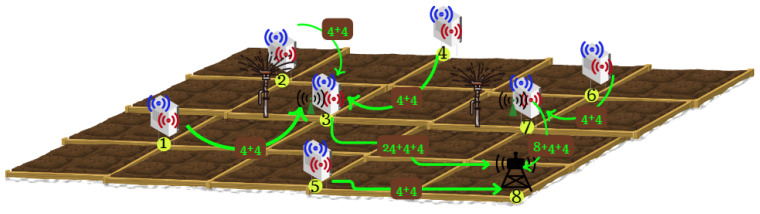
Solution of the OEO model.

**Figure 15 sensors-24-01457-f015:**
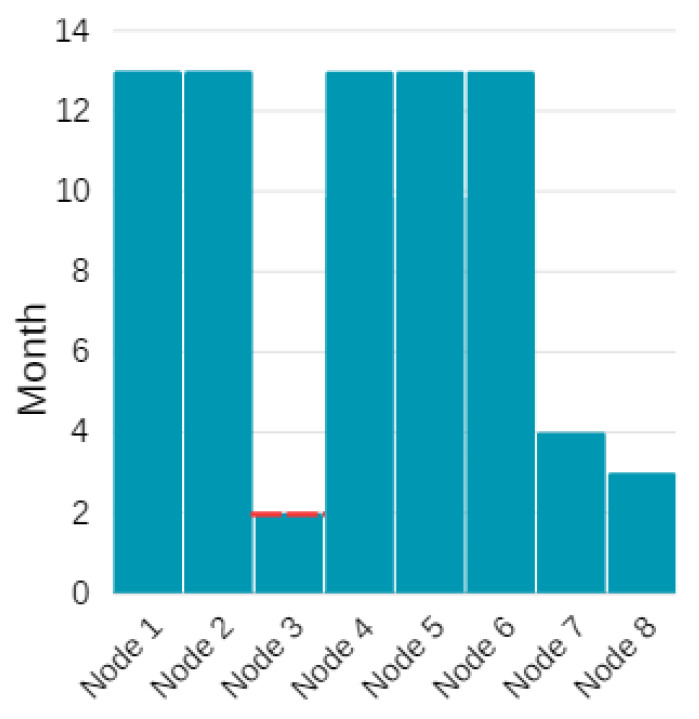
Lifetime of each node in terms of months.

**Table 1 sensors-24-01457-t001:** List of symbols.

*Index Sets*	
N	set of candidate locations
$K_{1}$	set of sensor types
$K_{2}$	set of node types
*Sensor Parameters*	
$\beta_{k}$	bandwidth of a type *k* sensor
$c_{k}$	cost of a type *k* sensor
$r_{k}^{c}$	communication radius of a type *k* sensor
$r_{k}^{s}$	sensing radius of a type *k* sensor
$a_{jkj^{\prime }}$	1 if point $j^{\prime }$ is within the sensing radius of a sensor (j,k), 0 otherwise
$b_{jkj^{\prime }}$	1 if point $j^{\prime }$ is within the communication radius of a sensor (j,k), 0 otherwise
$h_{jk}$	data packet size generated by a sensor (j,k) in an active period
$e_{k}^{c}$	energy consumed by a type *k* sensor to transmit one unit of data
$e_{k}^{ar}$	energy consumed by a type *k* sensor to receive one unit of data in an active period
$e_{k}^{sr}$	energy consumed by the type *k* sensor in a sleep period
$e_{k}^{a}$	energy consumed by a type *k* sensor in an active period
$e_{k}^{s}$	energy consumed by a type *k* sensor for sensing and data processing in an active period
$E_{j}$	initial battery energy of a node at point *j*
*General Parameters*	
${\bar{c}}_{j}$	cost of a node deployment at point *j*
*B*	available budget
${\bar{f}}_{ik}$	coverage requirement of a point *i* by a type *k* sensor without redundancy
$f_{ik}$	coverage requirement of a point *i* by a type *k* sensor with redundancy
α	minimum number of nodes to communicate with
*Decision Variables*	
$x_{jk}$	1 if a sensor (j,k) is deployed, 0 otherwise
${\bar{x}}_{j}$	1 if a gateway is deployed at point *j*, 0 otherwise
${\bar{y}}_{j}$	1 if there is a node at point *j*, 0 otherwise
$z_{jkt}$	1 if a sensor (j,k) is active in period *t*, 0 otherwise
$y_{iljkt}$	the amount of data sent from a sensor (i,l) to a sensor (j,k) in period *t*
$E_{jkt}$	energy consumption of a node (j,k) in period *t*
*L*	network lifetime

**Table 2 sensors-24-01457-t002:** Current consumption of the circuit during operation.

Operation	Sleep Current (mA)	Wake Current (mA)
Sensing and Transmitting	6	31
Only Transmitting	6	89
Only Receiving	6	21

**Table 3 sensors-24-01457-t003:** Cost of network elements.

Network Element	Cost (TL)
Gateway	935
Router	935
Temperature Sensor	480
Humidity Sensor	480

## Data Availability

Data are contained within the article.

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
