# Peer review of "A Mathematical Programming Approach for IoT-Enabled, Energy-Efficient Heterogeneous Wireless Sensor Network Design and Implementation"

_sensors, 2024, doi:10.3390/s24051457_

Round 1
Reviewer 1 Report
Comments and Suggestions for Authors
In this paper, the authors aim to place boxes containing various sensors on agricultural land to monitor it for different parameters. They introduce a mathematical model that minimizes the energy consumption of sensors in an agricultural field, reduces the number of boxes placed inside the sensors, and minimizes the number of hops to prevent data loss. They offer a mixed-integer programming solution to minimize the number of boxes. Heuristic methods, valid inequalities, and branch-and-bound methods are integrated in order to achieve optimal results within an acceptable time frame for real agricultural fields.
IMO, the proposed mathematical programming approach has merit and the considered topic is interesting. However, the main weaknesses of the paper are:
(1) In the following paper (not cited), the authors have presented a similar mathematical approach that deploys a minimum number of devices and efficiently utilizes the device battery by planning their active/sleep schedules and determining the minimum energy-consuming data flow paths. I consider that the authors must clearly state the differences between their proposed mathematical solution with their already published mathematical approach.
K. Olcay, E. Taparci, M. O. Akmandor, B. Kabakulak, B. Sarioglu and Y. D. Gokdel, "Modeling and Implementation of an Adaptive Wireless Sensor Network for Low Power IoT Applications," 2023 8th International Conference on Smart and Sustainable Technologies (SpliTech), Split/Bol, Croatia, 2023, pp. 1-4, doi: 10.23919/SpliTech58164.2023.10193646.
(2) In the Introduction section, the authors must clearly present the contribution of their mathematical model.
(3) The readability of the paper must be improved. To this end, the authors must improve the "flow" of the ideas presented in the whole paper. The paper must become easy to read and understood at first reading.
(4) The English language must be polished.
Comments on the Quality of English LanguageThe English language must be polished.
Author Response
It is attached.

Reviewer 2 Report
Comments and Suggestions for Authors
In this manuscript, the authors propose a mathematical model and its hardware implementation for the problem of distributing sensors on agricultural land and collecting data from the field in order to increase crop yield. Overall, the work presented has certain novelty and application value. However, the overall quality of the manuscript leaves considerable room for improvement, and the authors should consider addressing the following issues:
1. The title of the manuscript is not entirely consistent with the work presented and does not reflect the agricultural application context repeatedly mentioned in the manuscript.
2. The current abstract does not meet the requirements of a typical scientific research journal article. It is suggested that the authors make appropriate adjustments to the content of the abstract in order to present the motivations, methods, results, conclusions, and significance of their work in an appealing manner.
3. There are obvious problems with the handling of abbreviations.
4. The mathematical symbols in Figure 3 should be displayed in italics.
5. References, equations, figures, tables, chapters, etc. that appear in the manuscript should be added with hyperlinks to facilitate readers' reading.
6. There are serious problems with the representations and explanations of all equations and mathematical symbols.
7. At the beginning of each section, the authors should briefly introduce the content of the section in a few sentences.
8. The font size of some figures is too large, such as Figures 7 and 11.
9. There are inconsistencies in terminologies and descriptions, such as "equation (7)/Equation 26".
10. Equations 27 and 33 suffer from similar obvious problems.
11. The division of paragraphs is extremely arbitrary, and there are a large number of short paragraphs of two or three sentences in the manuscript.
12. The conclusion section obviously does not meet the requirements. The authors should present current problems, solutions, contributions, findings, limitations, and prospects for future work in a more engaging manner.
Comments on the Quality of English Language
Moderate editing of English language required.
Author Response
Thank you very much for taking the time to review this manuscript. You can see the modified parts as red coloured in the manuscript.
Reviewer 3 Report
Comments and Suggestions for Authors
The paper titled "A Mathematical Programming Approach for IoT-enabled Energy-Efficient Heterogeneous Wireless Sensor Network Design and Implementation" by Ertugrul Taparci et al. discusses the development of an energy-efficient sensor network for smart farming. It leverages IoT technologies and mathematical programming for optimal placement and energy management of sensor nodes.
My technical review comments are:
Strengthen the introduction with a clearer statement of the work's novelty and a more detailed comparison with existing technologies.
Elaborate on the assumptions and limitations of the mathematical models to provide a better understanding of their applicability.
Enhance the section on hardware implementation with more details on practical testing and validation, possibly including case studies or real-world deployment scenarios.
Provide a deeper analysis of the results, including comparisons with existing methods to demonstrate the effectiveness of the proposed system.
In the conclusion, discuss potential future improvements and other areas where this system could be applied, such as different agricultural settings or other IoT applications.
Author Response
Thank you very much for taking the time to review this manuscript. You can see the modified part as red coloured in the manuscript.
Round 2
Reviewer 1 Report
Comments and Suggestions for Authors
Please cite the previous work
K. Olcay, E. Taparci, M. O. Akmandor, B. Kabakulak, B. Sarioglu and Y. D. Gokdel, "Modeling and Implementation of an Adaptive Wireless Sensor Network for Low Power IoT Applications," 2023 8th International Conference on Smart and Sustainable Technologies (SpliTech), Split/Bol, Croatia, 2023, pp. 1-4, doi: 10.23919/SpliTech58164.2023.10193646.
Comments on the Quality of English LanguageMinor editing of English language required
Author Response
Thank you very much for taking the time to review this manuscript. We have updated Abstract, Introduction, Conclusion, and some part of the Sensor Network Design part according to the previous Reviewer's comments as well. These were the improvements points of the manuscript you declared. We cited our previous work accordingly in the introduction and Literature review part.